# Carrageenan of Red Algae *Eucheuma gelatinae*: Extraction, Antioxidant Activity, Rheology Characteristics, and Physicochemistry Characterization

**DOI:** 10.3390/molecules27041268

**Published:** 2022-02-14

**Authors:** Hoang Thai Ha, Dang Xuan Cuong, Le Huong Thuy, Pham Thanh Thuan, Dang Thi Thanh Tuyen, Vu Thi Mo, Dinh Huu Dong

**Affiliations:** 1Department of Food Technology, Ho Chi Minh City University of Food Industry, Ho Chi Minh 700000, Vietnam; haht@hufi.edu.vn (H.T.H.); dongdh@hufi.edu.vn (D.H.D.); 2Department of Biology, Graduate University of Science and Technology, VAST, Ha Noi 100000, Vietnam; thaonguyenxanh1607@gmail.com; 3Department of Organic Material from Marine Resource, Nha Trang Institute of Technology Research and Application, VAST, Nha Trang 650000, Vietnam; 4Institute of Biotechnology and Food Technology, Industrial University of Ho Chi Minh City, Ho Chi Minh 700000, Vietnam; 5General Surgery Department, Ninh Thuan Provincial General Hospital, Phan Rang 59000, Vietnam; phamthanhthuan2015@gmail.com; 6Department of Food Science, Nha Trang University, Nha Trang 650000, Vietnam; thanhtuyen151809@gmail.com

**Keywords:** *β*-carrageenan, antioxidant activity, Box-Behken, extraction, *Eucheuma gelatinae*, physic-chemistry, rheology

## Abstract

Carrageenan is an anionic sulfated polysaccharide that accounts for a high content of red seaweed *Eucheuma gelatinae*. This paper focused on the extraction, optimization, and evaluation of antioxidant activity, rheology characteristics, and physic-chemistry characterization of β-carrageenan from *Eucheuma gelatinae*. The extraction and the optimization of β-carrageenan were by the maceration-stirred method and the experimental model of Box-Behken. Antioxidant activity was evaluated to be the total antioxidant activity and reducing power activity. The rheology characteristics of carrageenan were measured to be gel strength and viscosity. Physic-chemistry characterization was determined, including the molecular weight, sugar composition, function groups, and crystal structure, through GCP, GC-FID, FTIR, and XRD. The results showed that carrageenan possessed antioxidant activity, had intrinsic viscosity and gel strength, corresponding to 263.02 cps and 487.5 g/cm^2^, respectively. Antioxidant carrageenan is composed of rhamnose, mannose, glucose, fucose, and xylose, with two molecular weight fractions of 2.635 × 10^6^ and 2.58 × 10^6^ g/mol, respectively. Antioxidant carrageenan did not exist in the crystal. The optimization condition of antioxidant carrageenan extraction was done at 82.35 °C for 115.35 min with a solvent-to-algae ratio of 36.42 (*v*/*w*). At the optimization condition, the extraction efficiency of carrageenan was predicted to be 87.56 ± 5.61 (%), the total antioxidant activity and reducing power activity were predicted to 71.95 ± 5.32 (mg ascorbic acid equivalent/g DW) and 89.84 ± 5.84 (mg FeSO_4_ equivalent/g DW), respectively. Purity carrageenan content got the highest value at 42.68 ± 2.37 (%, DW). Antioxidant carrageenan from *Eucheuma gelatinae* is of potential use in food and pharmaceuticals.

## 1. Introduction

*Eucheuma gelatinae* belongs to the *Solieriaceae* family, specifically the Rhodophyta division, and is a commonly popular marine plant acting as the material for processing β-carrageenan that is widely used in food, functional foods, and pharmaceuticals. The *Eucheuma gelatinae* species is a small individual size and lives in dead coral areas. Currently, the demand for carrageenan increases more in the commercial rhodophytes and plays a vital role in the world [1,2,3]. The *Eucheuma* species was the most farmed red algae, with 10.2 million tonnes in 2015, cultured in Korea, the Philippines, Malaysia, and China as the main algae species [4]. There were about 30,000 tons of cottonii (*Eucheuma alvarezii* Doty), 6000 tons of spinosum (*Eucheuma denticulatum* [Burman] Collins & Hervey), and 100 tons of gelatinae (*Eucheuma gelatinae* [Esper] J. Agardh) farmed for producing *κ*-carrageenan, *ι*-carrageenan, and a mixture of *γ*-, *β*- and *κ*-carrageenans [5], respectively.

Carrageenan is a galactan polysaccharidesand exists in the intercellular matrix of red algae. Carrageenan possesses numerous various bioactivities, for instance, anticoagulant [6], antiviral [7], antithrombotic [8], antibacterial [9], cholesterol-lowering [10], antitumor [11], immunomodulatory [12], and antihyperlipidemic [13]. It is also used in the treatment of stomach ulcers [14], and as an antioxidant. The bioactivity of carrageenan was clearly both in vitro and in vivo, and led to the potential promising in developing therapeutic agents. Among those biological activities, the antioxidant activity of carrageenan is most remarkable because antioxidants will eliminate free radicals and contribute to improving resistance and minimizing diseases in the human body. Therefore, carrageenan has potential in functional foods and pharmaceuticals, and is commonly applied in the food and pharmaceutical industries, and is used primarily for drug delivery (tablets, suppositories, fast-dissolving insert, beads, pellets, films, oral suspensions, micro/nanoparticles, floating model, intranasal system, wafers, hydrogel, and tissue engineering (bone or cartilage, and 3-D bioprinting applications) [15] in the latter. Most publications are mainly on *kappa* carrageenan from *Eucheuma denticulatum* and *Kappaphycus alverazii*.

Nowadays, there are numerous different methods for carrageenan extraction, for example, enzyme-assisted extraction [16], maceration [17], stirring soak [18], pressurized-assisted maceration [19], ultrasound-assisted [20], microwave-assisted [21] and extraction optimization [22]. The results on carrageenan from *Eucheuma gelatinae* were less and did not present the content, antioxidant activity, rheological and physicochemical properties of carrageenan from *Eucheuma gelatinae*, especially species grown in Vietnam. Carrageenan application development in functional foods and pharmaceuticals, the control of extraction conditions, the extraction optimization of multi-objective functions including refined carrageenan content, the antioxidant activity, rheology, and the physicochemical properties of carrageenan all demonstrate its essential role.

Therefore, the study focuses on the extraction and optimization of antioxidant carrageenan extracting from *Eucheuma gelatinae* grown in Vietnam, and the evaluation of its antioxidant activity, rheology characteristics, and physiochemistry properties.

## 2. Results

### 2.1. Extraction of Antioxidant Carrageenan

#### 2.1.1. Purity Carrageenan Content

The purity of carrageenan content varied from 23.41 ± 1.27 to 41.94 ± 3.05 (%, DW) in the range of the extraction condition as described in Section 4.3 (Table 1). Solvent (pH 7) caused the highest purity carrageenan content, compared to other pH solvents at the same condition. The significant difference in the purifying carrageenan content (*p* < 0.05) occurred as a pH solvent over 8. Pure carrageenan content was the highest at 100 °C, compared to others. However, using an extracting temperature from 80 to 100 °C affected non-significantly the pure carrageenan content (*p* > 0.05), except for the temperature, which lowered to 80 °C. Purity carrageenan content was in the range of 33.77 ± 1.55 to 39.62 ± 1.82 (%, DW) as surveying the extracting temperature. Pure carrageenan content got 32.45 ± 1.62 (%, DW) and 39.52 ± 2.11 (%, DW) at the extracting time of 30 min and 120 min, respectively. The difference in purity carrageenan content did not occur while the extraction time was from 60 to 120 min (*p* > 0.05) (Table 1). The highest purity carrageenan content was 41.02 ± 3.52 (%, DW) for the extracting time of 90 min. The solvent-to-algae ratio was significantly affected purity carrageenan content (*p* < 0.05) as lower than 30/1 (*v*/*w*). The purity carrageenan content of 42.68 ± 2.37 was found at the solvent-to-algae ratio of 40/1 (*v*/*w*), and this was the highest value compared to other conditions.

#### 2.1.2. Antioxidant Activity

##### Total Antioxidant Activity

The total antioxidant activity was in the range of 10.04 ± 1.68 to 30.76 ± 3.21 (mg ascorbic acid equivalent/g DW) when the survey of the extraction conditions and the change of total antioxidant activity were significant (*p* < 0.05). The difference in solvent pH led to the difference in total antioxidant activity (*p* < 0.05), except for solvents pH 8 and 9. The total antioxidant activity got the highest value of 24.58 ± 1.15 (mg ascorbic acid equivalent/g DW) at solvent pH 7 when compared to other solvents. The extracting temperature affected total antioxidant activity (*p* < 0.05) when the temperature increased from 70, 80 °C to 100 °C. The non-significant difference in total antioxidant activity occurred when the extracting temperature increased from 80 to 90 °C and 90 to 100 °C. However, the total antioxidant activity was still evaluated highest with the value of 29.31 ± 2.48 (Table 1).

The extracting time of carrageenan from 60 to 120 min did not significantly affect total antioxidant activity (*p* < 0.05); the extracting time of 30 min impacted to significant (*p* < 0.05), compared to other the extracting time. The highest total antioxidant activity was for 120 min compared to other extraction conditions.

Under the impact of the solvent-to-algae ratio, total antioxidant activity varied from 20.37 ± 1.92 to 30.04 ± 2.57 (mg ascorbic acid equivalent/g DW), corresponding to the solvent-to-algae ratio of 20/1 and 50/1 (*v*/*w*). However, the difference in total antioxidant activity only occurred (*p* < 0.05) when the solvent-to-algae ratio was lower than 30/1 (*v*/*w*) in comparison to other solvent-to-algae ratios.

##### Reducing Power Activity

Reducing power activity changed from 20.57 ± 2.03 to 36.25 ± 3.01 (mg FeSO_4_ equivalent/g DW) under the impact of other extraction conditions. Reducing power activity got the highest value of 27.84 ± 2.71 (mg FeSO_4_ equivalent/g DW) at solvent pH 9, but was not a significant difference from solvent pH 7 and 8. Reducing power activity was the lowest, corresponding to 20.61 ± 2.52 (mg FeSO_4_ equivalent/g DW) at solvent pH 10. The extracting temperature only led to a significant difference (*p* < 0.05) in reducing power activity when the temperature was below 80 °C, compared to other temperatures. The reducing power activity of 30.59 ± 3.26 (mg FeSO_4_ equivalent/g DW) was the highest value, compared to 70 to 90 °C. The reducing power activity (32.68 ± 3.24, mg FeSO_4_ equivalent/g DW) was highest for the extracting time of 120 min compared to other times (Table 1). The reducing power activity was not significantly different, compared at 60 and 90 min. The lowest reducing power activity (23.17 ± 1.78, mg FeSO_4_ equivalent/g DW) exhibited a significant difference (*p* < 0.05) in comparison to other extracting times.

Under the impact of the solvent-to-algae ratios, the reducing power activity varied from 25.73 ± 2.75 to 36.25 ± 3.01 (mg FeSO_4_ equivalent/g DW). A significant difference did not occur (*p* > 0.05) between the solvent-to-algae ratio of 20/1 and 30/1 (*v*/*w*), 30/1 and 40/1 (*v*/*w*), and 40/1 and 50/1 (*v*/*w*). The reducing power activity reached the highest and lowest value when the solvent-to-algae ratio was 50/1 and 20/1 (*v*/*w*), respectively.

#### 2.1.3. Correlation between Carrageenan Content and Antioxidant Activity

Purity carrageenan content strongly correlated to total antioxidant activity and weakly to reducing power activity, corresponding to 0.97 and 0.41 when the impact survey of solvent pH on purity carrageenan content and antioxidant activity was added. There is a strong correlation between purity carrageenan content and antioxidant activity (R^2^ > 0.9), especially total antioxidant activity (0.99), and reducing power activity (0.97) as with the extracting temperatures survey. The total antioxidant activity and reducing power activity strongly correlated to the purity carrageenan content, corresponding to 0.94 and 0.91, respectively, as the extracting time survey. A strong correlation was found between purity carrageenan content and antioxidant activity, as the survey of the solvent-to-algae ratio corresponded to 0.99 (total antioxidant activity) and 0.88 (reducing power activity), respectively.

### 2.2. Optimization of Antioxidant Carrageenan

#### 2.2.1. Analysis of Optimization Model

According to the study type of response surface with the design type of Box-Behnken on the quadratic model and randomized subtype, the results showed the distribution of the target functions were focusing on the centre of the survey interval, compared with the boundary region. The fit optimization model of *Y*_1_ function was the quadratic model (*p* = 0.0001 < 0.05) with non-significant lack-of-fit (*p* = 0.08 > 0.05) and adjusted R^2^ (0.95). Response surface *Y*_1_ had a standard deviation (SD) of 5.61 and coefficient of variation (C.V%) of 13.51. Response surface *Y*_2_ was the model quadratic (*p* = 0.0002 < 0.05), compared to the model of 2FI, linear, and cubic. SD and C.V% of *Y*_2_ were 2.53 and 15.8, respectively. The lack-of-fit of model *Y*_2_ (*p* = 0.43 > 0.05) was non-significant, and its adjusted R^2^ got 0.94. Sequential *p*-value and adjusted R^2^ of the response surface *Y*_3_ corresponded to the one of response surface *Y*_1_. The lack of fit of model *Y*_3_ had a *p*-value of 0.1. The quadratic model *Y*_3_ got a C.V% of 13.74 and SD of 2.53.

The response surface *Y*_1_ was in the range of 15.82 to 88.24 (%) and got the average value of 41.52 ± 5.61 (%). The response surface *Y*_2_ changed in the range of values (5.11 to 36.20 mg ascorbic acid equivalent/g DW), and its average value got 16.03 ± 2.53 (mg ascorbic acid equivalent/g DW). The value range of response surface *Y*_3_ varied from 5.94 to 39.53 (mg FeSO_4_ equivalent/g DW), with the average value of 18.43 ± 2.53 (mg FeSO_4_ equivalent/g DW) (Table 2).

After ANOVA analysis for three response models, the results showed the solvent-to-material ratio had a non-significant effect on all response surfaces because their *p*-value was 0.9527, 0.9606, and 0.9521, corresponding to surface *Y*_1_, *Y*_2_, and *Y*_3_, respectively. Interaction factors of *x*_1_*x*_2_ and *x*_2_*x*_3_ possessed a *p*-value higher than 0.05, showing that these interaction factors did not affect the response surfaces (Table 3). The results also showed the coding variable equation of response surface *Y*_1_, *Y*_2_, and *Y*_3_, as follows:*Y*_1_ = 74.17 + 37.02*x*_1_ − 44.98*x*_2_ − 0.1796*x*_3_ + 2.03*x*_1_*x*_2_ + 11.36*x*_1_*x*_3_ − 4.72*x*_2_*x*_3_ − 55.50*x*_1_^2^ − 70.28*x*_2_^2^ − 27.82*x*_3_^2^(1)
*Y*_2_ = 28.93 + 14.51*x*_1_ − 18.11*x*_2_ − 0.0675*x*_3_ + 0.7778*x*_1_*x*_2_ + 4.34*x*_1_*x*_3_ − 1.80*x*_2_*x*_3_ − 21.94*x*_1_^2^ − 28.00*x*_2_^2^ − 11.05*x*_3_^2^
(2)
*Y*_3_ = 32.94 + 16.44*x*_1_ − 20.04*x*_2_ − 0.0821*x*_3_ + 0.9111*x*_1_*x*_2_ + 5.04*x*_1_*x*_3_ − 2.10*x*_2_*x*_3_ − 24.65*x*_1_^2^ − 31.28*x*_2_^2^ − 12.37*x*_3_^2^(3)

The importance of response surfaces was equal, and the extraction optimization of antioxidant carrageenan via the software Design-Expert version 13 showed the optimization point of 82.35 °C, 115.35 min, and 36.42 (*v*/*w*) with the overlay figure of response surfaces (Figure 1d). Antioxidant carrageenan was the white color and yarn type in the optimization point (Figure 1e,f). At the optimization condition, response surfaces *Y_1_*, *Y_2_*, and *Y_3_* were predicted to get the average value of 87.56 ± 5.61 (%), 71.95 ± 5.32 (mg ascorbic acid equivalent/g DW), and 89.84 ± 5.84 (mg FeSO_4_ equivalent/g DW), respectively. Response surfaces were the spherical surface (Figure 1a–c).

#### 2.2.2. Test of Optimization Model by the Experiment

Following the experiment on the optimization condition and the correlation analysis between the actual target functions and the predicted target function, the results showed the strong correlation between the experiment and the prediction (Figure 2). The experiment value of the target functions corresponded to 86.52 (*Y*_1_) (Figure 2a), 87.69 (*Y*_2_) (Figure 2b), and 85.73% (*Y*_3_) (Figure 2c) when compared to the predicted target functions by the software Design Expert version 13.

### 2.3. Characteristics of Rheology and Physical-Chemistry of Antioxidant Carrageenan

#### 2.3.1. Rheological Characteristic of Antioxidant Carrageenan

Intrinsic viscosity and gel strength of antioxidant carrageenan had a value of 263.02 cps and 487.5 g/cm^2^, respectively.

#### 2.3.2. Physical-Chemistry Characteristics of Antioxidant Carrageenan

Different sugars were found in antioxidant carrageenan, such as rhamnose, mannose, glucose, fucose, and xylose. These sugars got the value of 59.16, 52.63, 78.20, 20.24, and 96.98, respectively. Galactose was not detected in antioxidant carrageenan (Figure 3).

The FTIR method utilizes the material’s light absorption by manipulating how different molecular compounds respond to infrared light to determine the analyzed material’s structure. This method is also known as absorption spectroscopy. It is applied in various ways, including light beams of a limited frequency group or using monochromatic light. This technique exploits that the fact that each frequency responds differently to the material and works by using more than one different frequency in the beam. In this way, the composition of unknown material is precisely determined. FTIR spectroscopy offers the advantage of measuring a small sample (a few milligrams or milliliters) in the least amount of time. FTIR analysis in the spectrum range of 580–3420 cm^−1^ showed different peaks, such as 3416.64, 2926.42, 1722.49, 1643.63, 1417.96, 1376.90, 1264.34, 1161.71, 1071.64, 845.86, and 582.00 cm^−1^ occurring in the FTIR of antioxidant carrageenan extracting from *Eucheuma gelatinae* grown in Vietnam. The things showed the functional groups, for example, -OH, C-H, C=O, NH_2_ deformation, C-O-H stretch or C-O/C-H bending, C-O or CH_3_ deformation, alkyl ketone or C-O-C stretch, alkylamine, sulphation of C4 of the /3–1,3-linked residue, and S-O-S bending (Appendix A). The peak of 842.89 cm^−1^ exhibited the C-O-SO_4_ group on C_4_ of galactose (Figure 4). The peak of 927.76 was the characteristic for *k*-carrageenan without *µ*-carrageenan and presented 3,6-anhydro-D-galactose group (Figure 4). The stretching vibration of the entire anhydro-glucose ring of antioxidant carrageenan was presented at the peak of 574.79 cm^−1^, 769.60 cm^−1^, 891.11 cm^−1^, and 927.76 cm^−1^ (Figure 4), respectively. The peak of 891.11 cm^−1^ and 842.89 cm^−1^ were the properties of *β*-carrageenan and *j*-carrageenan, respectively (Figure 4).

GPC is a new solution for the effortless analysis of polymeric compounds in nature and facilitates molecular mass analysis; carrageenan is dissolved in an alkaline medium. Figure 5 shows that antioxidant carrageenan, which was extracted from *Eucheuma gelatinae*, possessed two fractions, with the average molecular weight of 2.635 × 10^6^ and 2.58 × 10^6^ g/mol, respectively.

X-ray diffraction (XRD) is the sole laboratory technique that equips structural information such as chemical composition, crystal structure, crystal size, strain, and layer thickness. As a result, materials researchers use XRD to examine a wide range of materials, from powder X-ray diffraction (XRPD) to solids, thin films, and nanomaterials. In the current study, X-ray diffraction was used to analyze the crystal structure of carrageenan. Antioxidant carrageenan had a high purity degree and did not form a crystal structure, exhibited in Figure 6.

## 3. Discussion

Purity carrageenan content was collected using solvent pH 7 at 1.54 and 1.32 times, compared to solvent pH 10 and 9, respectively. The highest drop in purity carrageenan content occurred when solvent pH increased from 8 to 9, corresponding to 21.09%. In the range of the extracting temperature from 80–100 °C, the temperature increased by 10 °C, and purity carrageenan content increased by 6.8%. At the extracting temperature of 70 °C, the purity carrageenan content was 0.94, 0.88, and 0.85 times, compared to 80, 90, and 100 °C, respectively. The impact of solvent pH for purity carrageenan content was the descend linear model when solvent pH increased, however, one of the temperatures was inversed. The change of purity carrageenan content was the quadratic model trend with the maximum peak at 90 min. When the extracting time increased from 30 to 60 min, purity carrageenan content also increased by 18.89%. Increasing of the time from 60 to 90 min, the following increase of purity carrageenan content was only 6.32%. After the extraction at 90 min, purity carrageenan content decreased. The increase ratio of solvent-to-ratio from 20 to 40 (*v*/*w*) led to the increasing purity carrageenan content of 1.42 times. Purity carrageenan content tended parallel to the horizontal axis when the solvent-to-algae ratio increased from 30/1 to 50/1 (*v*/*w*). The difference in algae species and extraction methods caused different carrageenan-extracted content, for example, using ohmic heating for carrageenan extraction from *Eucheuma spinosum*, needing the temperature (95 °C), the time (240 min), and the solvent-to-algae ratio of 45/1 (*v*/*w*) [17]. The study of Andi et al. (2021) [17] exhibited the extraction condition of carrageenan from *Eucheuma spinosum* higher than in the current study.

The total antioxidant activity and reducing power activity changed according to the increasing solvent pH linear model trend, similar to purity carrageenan content under the impact of solvent pH. The increase of solvent pH was from 9 to 10, and total antioxidant activity and reducing power activity decreased by 55.58 and 35.08%, respectively. The antioxidant change of total antioxidant activity and reducing power activity was similar to purity carrageenan content change under the impact of the extracting temperature. It showed that antioxidant activity was proportional to the purity carrageenan content. The rate of increase in antioxidant activity decreased when the temperature increased. When the extracting temperature increased from 70 °C to 80, 90, and 100 °C, the ratio of total antioxidant activity increased by 27.23, 11.03, and 7.4%, the increasing ratio of reducing power activity corresponded to 30.04, 10.36, and 3.63%. The change of antioxidant activity composed of total antioxidant activity and reducing power activity was proportional to the purity carrageenan content change according to the linear model as the increase of extraction time. When the extracting time was 60, 90, and 120 min, total antioxidant activity increased by 43.26, 50.03, and 61.47%, and the increase of reducing power activity was 27.41, 29.86, and 41.05%, compared to the extracting time for 30 min. The impact of the solvent-to-algae ratio on purity carrageenan content and the antioxidant was similar to that of the extracting time. However, the effect of the raw material solvent ratio on the refined carrageenan content was only evident when changing the raw material solvent ratio from 20 to 30/1 (*v*/*w*). The solvent-to-algae ratio impacted antioxidant activity more clearly than the purity carrageenan content. It meant that carrageenan from *Eucheuma gelatinae* possessed antioxidant activity, and short-chain carrageenan had higher antioxidant activity than long-chain carrageenan. Previous studies had only shown the antioxidant capacity of *κ*-carrageenan. For carrageenan from *Eucheuma gelatinae*, its antioxidant activity was only shown based on the DPPH and ABTS method, and a correlation between carrageenan and antioxidant activity [23] was not found. In the current study, the antioxidant activity of beta carrageenan is evaluated based on total antioxidant activity and reducing power activity, also described as the correlation between them. The impact of different extraction conditions on antioxidant carrageenan from *Eucheuma gelatinae* was not presented in previous studies, but found in the current study.

Previous studies on optimization of carrageenan extraction mainly focused on the objective functions of extraction yield, gel strength, and viscosity of carrageenan on *Kappaphycus alverazii* and *Eucheuma spinosum* with alkaline solvent. The input factors were mainly studied to be temperature and time [24,25]. There was only one Chinese notice of *Eucheuma gelatinae* pretreatment optimization for carrageenan extraction. However, this publication did not address the antioxidant activity of carrageenan. The optimization of the target functions, such as extraction efficiency and antioxidant activity of carrageenan from *Eucheuma gelatinae*, the survey of solvent pH, temperature, time and solvent-to-material ratio in the determination of optimization domain, and the optimization of three input factors (solvent-to-material ratio, temperature, and time of extraction) in the optimization were presented in the current study. A Box-Behnken design model with a sphere response model had also not been found in studies of carrageenan extraction.

Antioxidant carrageenan from *Eucheuma gelatinae* in Vietnam possessed the characteristics of gel strength and viscosity higher than carrageenan from *Eucheuma sp.* in previous studies [24,26]. Sugar compositions of antioxidant carrageenan were noticed in the current study and were one of the few publications on sugar compositions of carrageenan and the only one on the sugar content of the antioxidant carrageenan from *Eucheuma gelatinae*. The average molecular weight of antioxidant carrageenan in the current study was higher than the results obtained by Nishinaria and Watase (1992) [27] and Ahmed et al. (2018) [28].

The results showed the existence of *β*-carrageenan in peak 891 cm^−1^ in the spectrum range of 580–3420 cm^−1^, [29], and the appearance of peak 820 cm^−1^ related to γ-carrageenan. The results were interesting compared to the previous studies. Algae species differently led in structure and functional groups composition in carrageenan [30], for example, *ι*-carrageenan in *Eucheuma serra* [31], and *κ*-carrageenan in *Eucheuma cottonii* [24]. The peak of 842 cm^−1^ and 925 cm^−1^ are C-O-SO_4_ group presence on C4 of galactose and 3,6-anhydro-D-galactose [18]. The entire anhydro-glucose ring achieved the stretching vibration at 573 cm^−1^, 760 cm^−1^, 858 cm^−1^, and 928 cm^−1^ [32], respectively. The spectra of 848 cm^−1^ and 891 cm^−1^ appeared as C-4 sulfate of the *j*-carrageenan and *β*-carrageenan, respectively [33].

## 4. Materials and Methods

### 4.1. Source of the Plant Material

Red algae *Eucheuma gelatinae* was collected from Ninh Hai district, Ninh Thuan province, Vietnam in April 2017.

### 4.2. Sample Preparation

*Eucheuma gelatinae* was selected, washed and dried until the humidity was lower than 8%. Next, the seaweed was crushed to a size of 2–3 cm for further study.

### 4.3. Extraction of Carrageenan

Carrageenan was extracted by agitating maceration with classical experimental design, fixing an independent variable and running the remaining variables.

For surveying solvent pH from 7 to 10 with a jump (δ) 1, temperature, time, and the ratio of solvent-to-algae of extraction were fixed, corresponding to 80 °C, 60 min, and 30/1 (*v*/*w*), respectively.

For surveying the extracting temperature from 70 to 100 °C with δ 10 °C, time and the ratio of solvent-to-algae of extraction were similar to the study on solvent pH and collection of solvent pH from the above results.

For surveying the extracting time from 30 to 120 min with δ 30 min, the ratio of solvent-to-algae of extraction was 30/1 (*v*/*w*). pH solvent and the extracting temperatures were from the above results.

For surveying the solvent-to-algae ratio from 20 to 50 (*v*/*w*) with δ 10 (*v*/*w*), three independent variables (solvent pH, temperature, and time of extraction) were from the above results.

All extracts were filtered through Whatman No 1. and precipitated using 96% ethanol. The residues were dried at 40 °C for carrageenan collection. Carrageenan was analysed for the purity content and antioxidant activity.

### 4.4. Optimization of Carrageenan Extraction

The optimization of carrageenan extraction was based on the experiment design model of Box-Behnken with four input factors, including temperature (*X*_1_, °C), time (*X*_2_, minutes), the solvent-to-material ratio (*X*_3_, *v*/*w*), and three target functions such as, for example, carrageenan purity degree (*Y*_1_, %), total antioxidant (*Y*_2_, mg ascorbic acid equivalent/g DW), and reducing power (*Y*_3_, mg FeSO_4_/g DW). The optimal experimental domain of four input factors and the independent-to-coding variable conversion are presented in Table 4. The experimental design included 12-factor experiments and three replicate experiments at the center of the plan (Table 5). The target functions were selected and based on the combined results of independent variables in Table 5. The experiments were randomly carried out to minimize the effects of unusual changes in the observations. The variables are coded according to the following equation:(4)$$x=\frac{\left(X_{i}-X_{0}\right)}{\Delta X}$$
wherein *x* is the code variable, *X_i_* is the real variable, *X*_0_ is the central experiment variable, and Δ*X* is the difference between the maximum value of the real variable and the value of *X*_0_. The mathematical equation corresponding to the Box-Behnken experimental model is as follows:(5)$$Y=\beta_{0}+\overset{3}{\underset{i=1}{\sum }}\beta_{i}X_{i}+\overset{3}{\underset{i=1}{\sum }}\beta_{ii}X_{i}^{2}+\overset{2}{\underset{i=1}{\sum }}\overset{3}{\underset{j=1+1}{\sum }}\beta_{ij}X_{i}X_{j}+\varepsilon$$

Requirement of objective functions:

*Y*_1_: extraction efficiency of carrageenan (%): max

*Y*_2_: total antioxidant activity (mg ascorbic acid equivalent/g DW): max

*Y*_3_: reducing power activity (mg FeSO_4_ equivalent/g DW): max

Carrageenan was collected in the condition of optimization extraction and analyzed for rheological (gel strength and intrinsic viscosity) and physical-chemistry characteristics (molecular weight, functional groups, crystal structure, and sugar composition).

### 4.5. Determination of the Content and the Extraction Efficiency of Carrageenan

#### 4.5.1. Purity Carrageenan Content

Purity carrageenan content is calculated based on the following equation:(6)$$Purity\;carrageenan\;content\;\left(C_{P},\;\%\right)=\frac{C_{R}-H_{C}-A_{C}}{W_{A}}\;\left(\%\right)$$
where in *C_R_* (%) is the dry weight of carrageenan after extraction as described in Section 4.3. The humidity content of *C_R_* is *H_c_*, which is determined according to the drying method at 105 °C until constant weight. *W_A_* is the dried algae powder. The ash content of *C_R_* is *A_c_*, which is calculated and based on the white ash weight of the material after being calcined at 650 °C.

#### 4.5.2. Extraction Efficiency of Carrageenan

The extraction efficiency of carrageenan was determined according to the following equation:(7)$$EEC\;\left(\%\right)=\frac{C_{p}}{Carrageenan\;in\;initial\;algae}\;\left(\%\right)$$
(8)$$Carrageenan\;in\;initial\;algae=W_{A}-H_{A}-A_{A}-Protein_{A}-Lipid_{A}-Cellulose_{A}$$
where in:

*W_A_*: the dried algae powder (g);

*H_A_*: humidity content of algae powder (g); dried at 105 °C.

*A_A_*: ash content of algae powder (g); calcined at 650 °C.

*Protein_A_*: protein content of algae powder (g); determined based on the method of Lowry.

*Lipid_A_*: lipid content of algae powder (g); determined based on the soxhlet method.

*Cellulose_A_*: cellulose content of algae powder (g);

### 4.6. Detemination of Antioxidant Activity

#### 4.6.1. Total Antioxidant Activity

100 µL extract, in turn, was added to 900 µL of distilled water and solution A (0.6 M H_2_SO_4_, 28 mM sodium phosphate and 04 mM ammonium molybdate). The mixture was vortexed and kept for 90 min at 95 °C and then measured at the wavelength of 695 nm with an ascorbic acid standard [34].

#### 4.6.2. Reducing Power Activity

The reducing power activity was determined according to the method of Zhu et al. (2002) [35]. Firstly, 0.5 mL phosphate buffer at pH 7.2 was added to 500 μL extract. Secondly, 0.2 mL of 1% K_3_[Fe(CN)_6_] was added to the compound. The compound was kept at 50 °C for 20 min. Thirdly, 500 μL of 10% CCl_3_COOH with 300 μL distilled water and 80 μL of 0.1% FeCl_3_ were added. Finally, the compound was measured at 655 nm with the standard substance FeSO_4_.

### 4.7. Determination of Rheological Characteristics

#### 4.7.1. Gell Strength

Carrageenan gel strength was measured with a Brookfield rheometer and the maintenance of samples at 20 °C. Samples were prepared by dissolving 1.7 g of carrageenan powder in 98.3 mL of distilled water at 80 °C to form a 1.5% carrageenan solution. KCl was then added until reaching 0.1% KCl solution and soaking to 20 °C for 2.5 h. Carrageenan gel was continuously cut into slices with a thickness of 1.5 cm and put into the rheometer for measurement.

#### 4.7.2. Intrinsic Viscosity

The viscosity of 1.5% carrageenan solution at 80 °C was measured with a Brookfield rheometer.

### 4.8. Determination of Physic-Chemistry Characterization

#### 4.8.1. Sugar Compositions

The sugar composition determination of antioxidant carrageenan was according to the GC-FID method on Agilent’s 6890 N gas chromatograph (USA) that was composed of an automatic sample injector, an injection chamber, a column furnace, a flame ionization detector (FID), and an HP5 MS column (30 m × 0.25 m × 0.25 m). The chamber temperature was set at 280 °C with the line split ratio of 0/1. The program temperature column was set at 100 °C with a 20 °C/min rate for getting 325 °C and kept for 10 min, and then the probe temperature at 300 °C and carried gas speed at 01 mL/min. Derivation process: The sample was hydrolyzed in 1.5 M HCl, then poured to 10 mL of the cylinder by acetic anhydride, and the compound was finally injected into the GC system.

#### 4.8.2. Molecular Weight

The average molecular weight of carrageenan was measured with gel permeation chromatography, using a model YL9100 GPC.

#### 4.8.3. Functional Groups

The samples and anhydrous KBr were mixed according to the ratio of KBr-to-sample of 98:2 (*w*/*w*). The mixture was then measured on a Bruker FT-IR spectrometer ALPHA with a wavelength range from 4000 to 500 cm^−1^. Finally, the results were analysed on OPUS 7.0 software (Bruker, Ettlingen, Germany).

#### 4.8.4. Crystal Structure

The crystal structure of carrageenan was measured on a Brucker-Germany instrument, model D8 Advance, which met the ISO 9001:2000 international standards, in addition to standards for radiation safety and European CE standards for electrical safety. 0.5 g of carrageenan was finely ground in an agate mortar and pestle and placed in a special tray of a polycrystalline powder diffraction apparatus. Samples in the tray were flat and spread evenly over the tray surface. The sample holder mounted on an incident beam (narrow, monochromatic, parallel X-ray beam projected onto the sample). X-ray was rotated at an angle theta (θ) for the incident ray, and obtained the dispersion of X-ray by the detector–right. The sodium iodide (NaI) flicker detector would rotate 2θ from 5–70°. One sample rotation was 0.03° and the single point diffraction time was one second. The measuring temperature of samples was at 25 °C.

### 4.9. Statistical Analysis

All experiments were undertaken in triplicate and exhibited under mean ± SD with a significant level (*p* < 0.05). Analysis of statistics, ANOVA, and regression was calculated using the software MS. Excel 2013 and Design Expert 13. Duncan method was used for the movement of the non-normal value.

## 5. Conclusions

In summary, surveying and condition optimization of carrageenan extraction from *Eucheuma gelatinae* with the target functions such as purity carrageenan content, total antioxidant activity, and reducing power activity were performed in this study. Antioxidant carrageenan extracted at the optimal condition was evaluated, including the rheology and physicochemical properties. Antioxidant carrageenan contained rhamnose, mannose, glucose, fucose, and xylose. The molecular weight of carrageenan reached an average value of 2.635e^6^ and 2.58e^6^ g/mol. The solvent-to-algae ratio had the least effect on the objective functions. Antioxidant carrageenan from *Eucheuma gelatinae* had intrinsic viscosity (263.02 cps) and gel strength (487.5 g/cm^2^), and did not exist in the crystal. At the optimization condition (82.35 °C for 115.35 min with the solvent-to-algae ratio of 36.42 (*v*/*w*)), target functions were predicted such as carrageenan yield of extraction (87.56 ± 5.61, %), total antioxidant activity (71.95 ± 5.32, mg ascorbic acid equivalent/g DW), and reducing power activity (89.84 ± 5.84, mg FeSO_4_ equivalent/g DW)). The highest value of purity carrageenan content is 42.68 ± 2.37 (%, DW), and it has potential in the food and pharmaceutical industries.

## Figures and Tables

**Figure 1 molecules-27-01268-f001:**
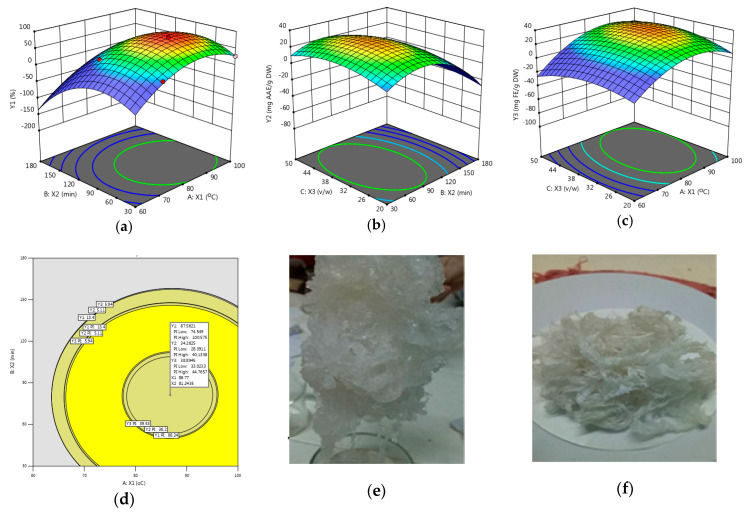
Response surface, overlay surface and antioxidant carrageenan: (**a**) Response surface of *Y*_1_; (**b**) Response surface of *Y*_2_; (**c**) Response surface of *Y*_3_; (**d**) Overlay surface of *Y*_1_, *Y*_2_, and *Y*_3_; (**e**) Carrageenan after precipitation using 96% ethanol; (**f**) Carrageenan after drying.

**Figure 2 molecules-27-01268-f002:**
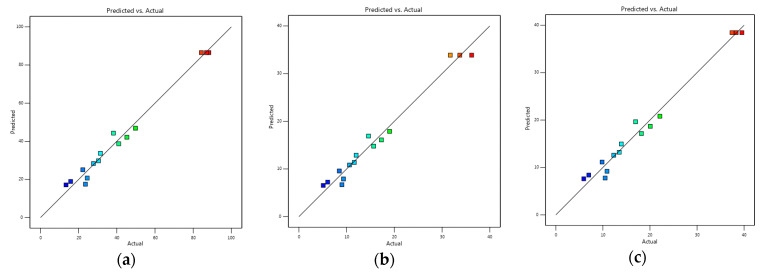
The correlation between predicted target function and actual target function: (**a**) *Y*_1_, (**b**) *Y*_2_, (**c**) *Y*_3_, respectively.

**Figure 3 molecules-27-01268-f003:**
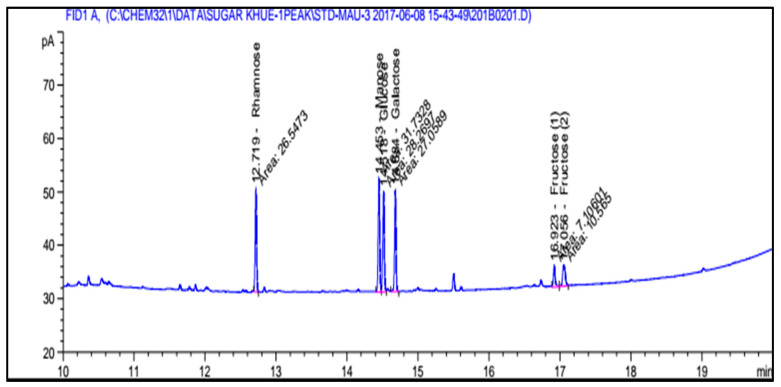
Sugar composition of antioxidant carrageenan.

**Figure 4 molecules-27-01268-f004:**
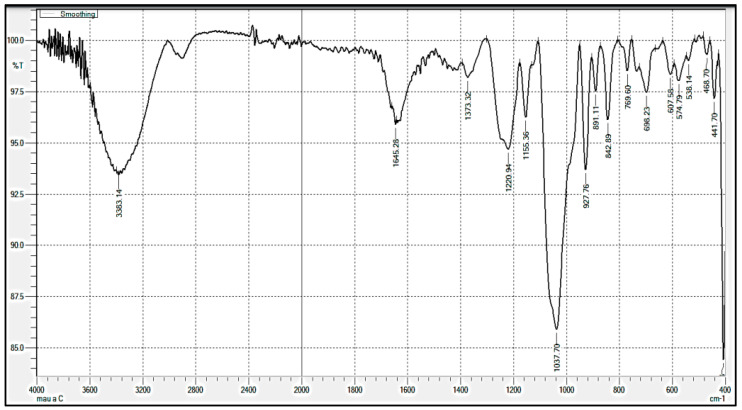
FTIR spectroscopy of antioxidant carrageenan.

**Figure 5 molecules-27-01268-f005:**
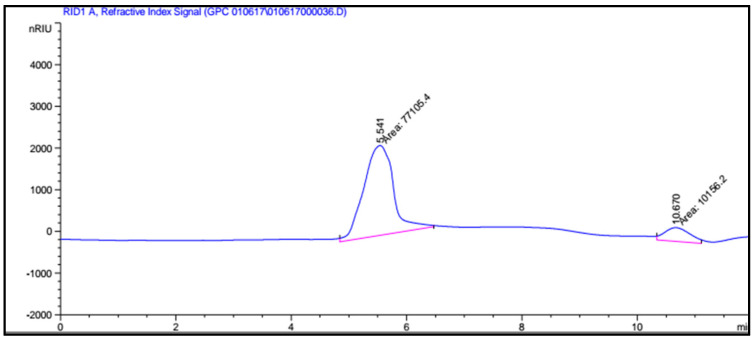
GPC spectroscopy of antioxidant carrageenan.

**Figure 6 molecules-27-01268-f006:**
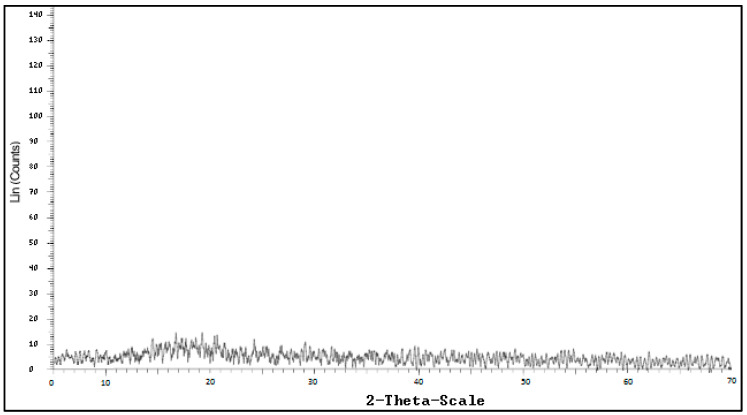
GPC spectroscopy of antioxidant carrageenan.

**Table 1 molecules-27-01268-t001:** Effect of the extraction condition on purity carrageenan content and its antioxidant activity.

Std	pH	Extracting Temperature (°C)	Extracting Time (min)	Solvent-to-algae Ratio (*v*/*w*)	Purity Carrageenan Content (%, DW)	Total Antioxidant Activity (mg Ascorbic Acid Equivalent/g DW)	Reducing Power Activity (mg FeSO_4_ Equivalent/g DW)
I	7	80	60	30/1	36.09 ± 1.21 ^a^	24.58 ± 1.15 ^a^	26.75 ± 1.98 ^a^
	8	80	60	30/1	34.57 ± 1.42 ^a^	19.73 ± 2.01 ^b^	23.29 ± 2.44 ^ac^
	9	80	60	30/1	27.28 ± 1.03 ^b^	15.62 ± 2.19 ^b^	27.84 ± 2.71 ^a^
	10	80	60	30/1	23.41 ± 1.27 ^c^	10.04 ± 1.68 ^c^	20.61 ± 2.52 ^bc^
II	7	70	60	30/1	33.77 ± 1.55 ^a^	19.32 ± 1.92 ^a^	20.57 ± 2.03 ^a^
	7	80	60	30/1	36.09 ± 1.47 ^ab^	24.58 ± 1.15 ^b^	26.75 ± 1.98 ^b^
	7	90	60	30/1	38.58 ± 1.69 ^b^	27.29 ± 2.75 ^bc^	29.52 ± 3.01 ^b^
	7	100	60	30/1	39.62 ± 1.82 ^bc^	29.31 ± 2.48 ^c^	30.59 ± 3.26 ^b^
III	7	90	30	30/1	32.45 ± 1.62 ^a^	19.05 ± 2.23 ^a^	23.17 ± 1.78 ^a^
	7	90	60	30/1	38.58 ± 1.69 ^b^	27.29 ± 2.75 ^b^	29.52 ± 3.01 ^b^
	7	90	90	30/1	41.02 ± 3.52 ^b^	28.58 ± 3.10 ^b^	30.09 ± 2.67 ^b^
	7	90	120	30/1	39.52 ± 2.11 ^b^	30.76 ± 3.21 ^b^	32.68 ± 3.24 ^b^
IV	7	90	90	20/1	30.14 ± 2.38 ^a^	20.37 ± 1.92 ^a^	25.73 ± 2.75 ^a^
	7	90	90	30/1	41.02 ± 3.52 ^b^	28.58 ± 3.10 ^b^	30.09 ± 2.67 ^ab^
	7	90	90	40/1	42.68 ± 2.37 ^b^	29.72 ± 3.22 ^b^	34.67 ± 2.78 ^bc^
	7	90	90	50/1	41.94 ± 3.05 ^b^	30.04 ± 2.57 ^b^	36.25 ± 3.01 ^cd^

Note: Std I, II, III, and IV included four lines. Letters a, b, c, and d in each column exhibited a significant difference in the column of each Std with *p* < 0.05, *n* = 3.

**Table 2 molecules-27-01268-t002:** The experiment results of the optimization design according to the Box-Behnken model.

Std	*X* _1_	*X* _2_	*X* _3_	*Y* _1_	*Y* _2_	*Y* _3_
1	70	30	35	23.56	9.02	10.48
2	100	30	35	27.74	10.60	12.30
3	70	120	35	30.40	11.61	13.48
4	100	120	35	38.24	14.59	16.94
5	70	75	20	22.18	8.47	9.83
6	100	75	20	24.49	9.35	10.87
7	70	75	50	13.40	5.11	5.94
8	100	75	50	49.79	19.01	22.11
9	85	30	20	15.82	6.04	7.00
10	85	120	20	41.01	15.66	18.19
11	85	30	50	31.42	12.00	13.92
12	85	120	50	45.28	17.30	20.07
13	85	75	35	88.24	33.71	37.43
14	85	75	35	86.97	31.72	39.53
15	85	75	35	84.30	36.20	38.32

Note: Values expressed as mean value, *n* = 3.

**Table 3 molecules-27-01268-t003:** The basic parameters of the response surface equation.

Source	Response Surface *Y*_1_			Response Surface *Y*_2_			Response Surface *Y*_3_		
	*p*-Value	CE	SE	*p*-Value	CE	SE	*p*-Value	CE	SE
Model	0.0007	74.17	3.02	0.0014	28.93	1.36	0.0008	32.94	1.36
*x*_1_ − *X*_1_	0.0004	37.02	4.47	0.0008	14.51	2.02	0.0004	16.44	2.02
*x*_2_ − *X*_2_	0.0018	−44.98	7.45	0.0030	−18.11	3.36	0.0019	−20.04	3.36
*x*_3_ − *X*_3_	0.9527	−0.1796	2.88	0.9606	−0.0675	1.30	0.9521	−0.0821	1.30
*x* _1_ *x* _2_	0.7575	2.03	6.23	0.7932	0.7778	2.81	0.7591	0.9111	2.81
*x* _1_ *x* _3_	0.0288	11.36	3.74	0.0499	4.34	1.69	0.0305	5.04	1.69
*x* _2_ *x* _3_	0.3590	−4.72	4.68	0.4325	−1.80	2.11	0.3652	−2.10	2.11
*x* _1_ ^2^	0.0001	−55.50	5.19	0.0002	−21.94	2.34	0.0001	−24.65	2.34
*x* _2_ ^2^	0.0003	−70.28	8.11	0.0006	−28.00	3.66	0.0004	−31.28	3.66
*x* _3_ ^2^	0.0002	−27.82	2.92	0.0004	−11.05	1.32	0.0002	−12.37	1.32

SE: Standard error; CE: Coefficient estimate.

**Table 4 molecules-27-01268-t004:** The conversion between code variables and reality variables.

Input Factor (Independent Variable)	Code Variable		
	−1	0	1
Extracting temperature (*X*_1_, °C)	70	85	100
Extracting time (*X*_2_, minutes)	30	75	120
Solvent-to-material ratio (*X*_3_, *v*/*w*)	20	35	50

**Table 5 molecules-27-01268-t005:** Experiment design and the results.

Std	Actual Variable			Coded Variable			Target Function		
	*X* _1_	*X* _2_	*X* _3_	*x* _1_	*x* _2_	*x* _3_	*Y* _1_	*Y* _2_	*Y* _3_
1	70	30	35	−1	−1	0	*Y_ij_*		
2	100	30	35	+1	−1	0			
3	70	120	35	−1	+1	0			
4	100	120	35	+1	+1	0			
5	70	75	20	−1	0	−1			
6	100	75	20	+1	0	−1			
7	70	75	50	−1	0	+1			
8	100	75	50	+1	0	+1			
9	85	30	20	0	−1	−1			
10	85	120	20	0	+1	−1			
11	85	30	50	0	−1	+1			
12	85	120	50	0	+1	+1			
13	85	75	35	0	0	0			
14	85	75	35	0	0	0			
15	85	75	35	0	0	0			

## Data Availability

Datas are available from the authors.

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
