# Peer review of "Carrageenan of Red Algae Eucheuma gelatinae: Extraction, Antioxidant Activity, Rheology Characteristics, and Physicochemistry Characterization"

_molecules, 2022, doi:10.3390/molecules27041268_

Round 1
Reviewer 1 Report
The manuscript is interesting to readers. The authors have used appropriate research design and characterization techniques. However, the authors must improve quality of present both in terms of quality of figures, currently the authors have used whatever they got from the instruments with marks. Images are very low in resolution. Also, have to present different characterization techniques in different subsections under result section and describe in little more detail.
English in the text needs to be improved.
they have to have a separate sub sections of different characterization techniques in
Author Response
Dear
The reviewer
Thankful for your review and help
The authors improved the quality of figures and languages
The author has also arranged and discussed each technique separately
The edits are highlighted in red by the author
Thank you very much
Best regards!

Reviewer 2 Report
This manuscript describes the extraction the optimization of carrageenan from Eucheuma gelatinae. Although the subject is interesting, the document has serious flaws, is very confusing and is not well structured. I don’t recommend this manuscript for publication since don’t meet the standards of Molecules Journal.
The document is not well written and an extensive editing of English language is required.
Author Response
Dear
The reviewer
Thankful for your review and help
The authors improved the quality of languages and the manuscript according to the comments of the reviewer
Thank you very much for your help
Best regards!

Reviewer 3 Report
The paper reports some interesting finding within the journal scoop however the current version seems bellow the journal standard.
The paper should be revised for English, grammar, and syntax errors
In the title, please correct “physic-chemistry” to “physicochemical characterization”
Line 64: in vivo in vitro should be in italic
Line 73: please correct to pharmaceuticals
The last paragraph in the introduction reporting the aims of this study should be rewritten and improved.
In the text do not write the letter “c” for ANOVA test (ex. 23.41±1.27c). please correct in whole manuscript.
Line 105: “…range of 10.04 ± 1.68 c to 30.76 ± 3.21 b when” the unit is missed
Line 114: please correct incomplete sentence: “did not cause significantly for…”
Figures 3, 4, 5 and 6 are of low quality and should be replaced by high resolution figures.
Equations should be numbered.
The conclusion is summarizing the main results without showing the key novelty and of this study; Please improuve.
Author Response
Dear
Thankful for your review and help
The authors revised the manuscript according to the comments of the reviewer
The parts highlighted in red are the parts that have been edited according to comments
Thank you very much for your help
Best regards!

Round 2
Reviewer 2 Report
The manuscript was improved, and essential information was added. However, the introduction the abstract and introduction need to be rewritten and improved. The information is not well organized and is very difficult to read. Moreover, the entire document needs an extensive editing of English language. In this sense, I recommend minor corrections before the manuscript is accepted for publication.
Author Response
Dear
The reviewer
Thankful for your help and review
The authors revised the manuscript according to the comments of the reviewer.
Corrections in the manuscript according to the reviewer's second comment are highlighted in red.
The authors hope to receive your good news and reply soon
Best regards!

Reviewer 3 Report
The paper is well revised according to previous comment. The current version seems within the standards.
No further correction required.
Author Response
Dear
The reviewer
Thankful for your help very much
Best regards!